# What risk do *Brucella* vaccines pose to humans? A systematic review of the scientific literature on occupational exposure

**Manuel Vives-Soto**[1], **Amparo Puerta-García**[1‡], **Esteban Rodríguez-Sánchez**[2‡], **José-Luis Pereira**[2‡], **Javier Solera**[3]*

**1** Albacete Quironsalud Hospital, Albacete, Spain, **2** CZ Vaccines Zendal Group, Pontevedra, Spain, **3** University of Castilla–La Mancha, Albacete, Spain

☉ These authors contributed equally to this work.
‡ AP-G, ER-S and J-LP also contributed equally to this work.
* solera53@gmail.com

## Abstract

### Background

Currently, vaccination of livestock with attenuated strains of *Brucella* remains an essential measure for controlling brucellosis, although these vaccines may be dangerous to humans. The aim of this study was to review the risk posed to humans by occupational exposure to vaccine strains and the measures that should be implemented to minimize this risk.

### Methods

This article reviewed the scientific literature indexed in PubMed up to September 30, 2023, following "the PRISMA guidelines". Special emphasis was placed on the vaccine strain used and the route of exposure. Non-occupational exposure to vaccine strains, intentional human inoculation, publications on exposure to wild strains, and secondary scientific sources were excluded from the study.

### Results

Nineteen primary reports were found and classified in three subgroups: safety accidents in vaccine factories that led to an outbreak (n = 2), survellaince studies on vaccine manufacturing workers with a serologic diagnosis of *Brucella* infection (n = 3), and publications of infection by vaccine strains during their administration, including case reports, records of occupational accidents and investigations of outbreaks in vaccination campaigns (n = 14). Although accidental exposure during vaccine manufacturing were uncommon, they could provoke large outbreaks through airborne spread with risk of spread to the neighboring population. Besides, despite strict protection measures, a percentage of vaccine manufacturing workers developed positive *Brucella* serology without clinical infection. The most frequent type of exposure with symptomatic infection was needle injury during vaccine administration. Prolonged contact with the pathogen, lack of information and a low adherence to

**Data Availability Statement:** The authors confirm that all data underlying the findings are fully

available without restriction. All relevant data are within the paper.

**Funding:** The author(s) received no specific funding for this work. The authors thank the Biofabri company for paying the costs of publishing this scientific article.

**Competing interests:** All authors declare that not competing interests exists.

personal protective equipment (PPE) use in the work environment were commonly associated with infection.

## Conclusions

*Brucella* vaccines pose occupational risk of contagion to humans from their production to their administration to livestock, although morbidity is low and deaths were not reported. Recommended protective measures and active surveillance of exposed workers appeared to reduce this risk. It would be advisable to carry out observational studies and/or systematic registries using solid diagnostic criteria.

## Author summary

Vaccination of livestock with attenuated strains of *Brucella* is an effective measure for controlling brucellosis, and they will continue to apply. Following "the PRISMA guidelines" we reviewed the risk posed to humans by occupational exposure to these strains and the measures that should be implemented to minimize this risk. Nineteen primary reports were included. The most frequent type of exposure was needle injury during vaccine administration, while safety accidents during vaccine manufacturing were less frequent but caused large outbreaks. Prolonged contact with the pathogen, lack of information and a low adherence to personal protective equipment (PPE) use in the work environment were commonly associated with infection. Despite strict protection measures, a percentage of vaccine manufacturing workers developed a positive serology to the vaccine strain without clinical infection. To conclude, *Brucella* vaccines pose risk of contagion to humans from their production to their administration to livestock, but with a low morbi-mortality.

## Introduction

Human brucellosis is a zoonosis with worldwide distribution. A recent study has estimated that there are at least 1.6–2.1 million new cases of human brucellosis each year [1]. Although the number of reported cases have decreased in more developed countries, the continued presence of the disease in some endemic areas, particularly in Eastern Europe, the Asia-Pacific, Central and South America, and Africa, and the potential use of *Brucella* species as an agent of bioterrorism, make brucellosis a major public health hazard with important sanitary and economic repercussions [2].

Currently, livestock vaccination remains an essential measure for the control of this zoonosis, and only live attenuated vaccines have shown efficacy in preventing infection in these animals [3,4]. The strains currently used in most countries for the control of bovine brucellosis are *Brucella abortus* S19 and *B. abortus* RB51, while the strain used for small ruminants is *B. melitensis* Rev.1 [5]. In China, *B. abortus* A19, a strain derived from S19, is used for cattle, and *B. abortus* S2 strain for pigs [6].

These strains are capable of establishing limited infection in livestock, mimicking the natural infection process by wild strains and thus conferring protection. However, these vaccines are not used in humans due to the high risk of developing acute brucellosis [4,7]; they are capable of infecting humans with occupational exposure via the oral, nasal, or conjunctival routes, and by accidental needle inoculation [8,9]. Whatever the route of entry, the infection can be

symptomatic or asymptomatic, and localized or systemic [9,10]. Serologic tests continue to be used to diagnose brucellosis, given the risk and difficulty involved in obtaining positive cultures [9,11]. However, serology cannot distinguish between vaccine and wild strains and current standard serologic assays cannot detect antibodies against RB51 infection [12]. The recent development of molecular techniques (PCR) has made it possible to distinguish vaccine from wild strains in animal and human samples [13].

Two meta-analyses on the occupational risks of contracting brucellosis have been published, with some references to the risks that *Brucella* vaccine handling poses for humans. One of these, by Xie et al (2018), analyzed 27 papers reviewing adverse effects in animals (n = 23) and humans (n = 4) associated with three licensed brucellosis vaccines: S19, Rev.1 and RB51 [14]. They found that human adverse effects from occupational exposure to the vaccines usually involved behavioral and neurological systems, and highlighted that no fatal or permanent human damage was reported. The other systematic review was published by Pereira et al. in 2020 [15], addressing the risk for human of contracting brucellosis by savage and vaccine *Brucella* strains. Regarding *Brucella* vaccine strains, they found only 7 elegible reports unsuitable for conducting a meta-analysis. New evidence has recently been published that sheds light on the risk of infection to humans by these vaccine strains.

At present, uncertainty remains about the real risk of these live attenuated vaccines for those who manufacture or administer them. Despite advances in the protection measures that must be applied when there is a risk of contact with these live bacterial vaccines [16], accidental contagion still occurs. The aim of this study was to review the risk posed to humans by occupational exposure to vaccine strains and the effectiveness of the preventive measures being implemented to minimize this risk.

## Material and methods

The guidelines of the PRISMA statement (Preferred Reported Items for Systematic Reviews and Meta-Analyses) were formally adopted in this review [17]. The search was conducted on September 30, 2023, with no date or country restrictions, using the PubMed database. Next, titles, abstracts and full texts were independently analyzed by two investigators. Three search algorithms were applied consecutively: firstly "*Brucella* AND (Rev-1 OR Rev.1 OR S19 OR S.19 OR B19 OR B-19 OR RB51 OR A19 OR S2) AND Human[Mesh]", secondly "*Brucella* AND Vaccine AND occupational AND Human[Mesh]", and thirdly "("Laboratory Infection"[Mesh]) AND *Brucella*[Mesh]". Once the relevant studies were selected, their references were reviewed. Non-occupational exposure to vaccine strains, intentional human inoculation, publications on exposure to wild strains, and secondary scientific sources were excluded from the study.

## Results

The PRISMA Flow Diagram is shown in Fig 1. Nineteen articles were included from 185 records screened. Of the articles finally included, 12 were identified with the first algorithm, 3 with the second algorithm, 1 with the third algorithm, and 3 more from review of the reference lists. Table 1 lists the publications of brucellosis cases acquired through laboral exposure to vaccine strains. These publications are summarized below in three subgroups: outbreaks after safety accidents in vaccine factories, risk for vaccine manufacturing workers, and risk during vaccine administration.

### Outbreaks after safety accidents in vaccine factories

Two safety accidents have been reported in *Brucella* vaccine factories. In both cases, a failure in the ventilation system and inadequate disinfection led to an outbreak through airborne

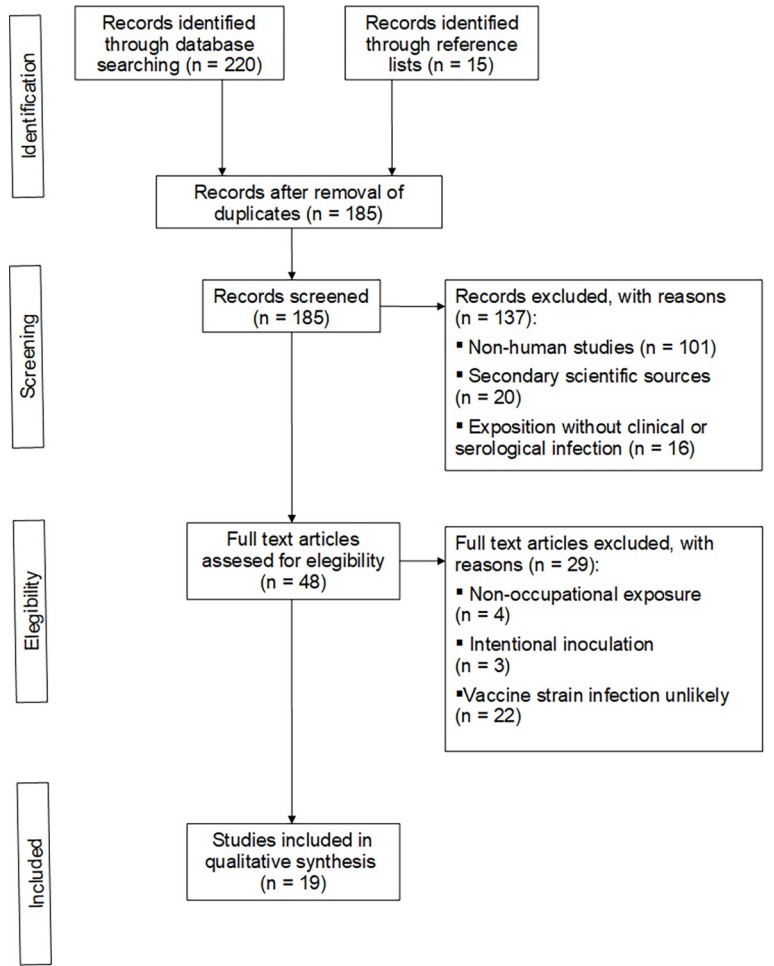

**Fig 1. PRISMA flow diagram.**

spread. Although the first one only affected factory workers, the second one in addition spread to the nearby general population, giving rise to a serious public health problem. These reports are briefly described below.

In 1987, Ollé-Goig et al. described an airborne-acquired outbreak of brucellosis in workers accidentally exposed to the Rev.1 strain at a manufacturing plant for veterinary biologic products in Gerona (Spain) [18]. The study included 164 workers, of which 22 had clinical symptoms and serology compatible with acute brucellosis, and six had "acute" serology without symptoms (attack rate: 17.1 per cent). On the other hand, 20 workers had chronic brucellosis, 106 were infection-free, and 18 had no clear diagnosis. The laboratory was located in a Spanish province not considered an area of especially high endemicity for brucellosis, making acute brucellosis acquired outside the factory very unlikely. The epidemiological research demonstrated that a failure in the ventilation system resulted in an outbreak of brucellosis in nearby workers.

During late July to August 2019, a laboratory accident with dramatic public health consequences occurred at a *Brucella* vaccine factory in the city of Lanzhou, located in Gansu province of northwest China. The outbreak initially affected 213 individuals from the nearby Veterinary Research Institute, 8 workers from the biopharmaceutical plant, 2,500 residents of neighboring areas, and 150 people located further away [19]. Until November 30, 2020, when the investigation was completed, the infection was confirmed in 10,528 individuals after

**Table 1. Publications of brucellosis cases acquired through exposure to vaccine strains.**

| Reference | Year | Country | Vaccine strain | Case / exposed | Diagnostic criteria |
|---|---|---|---|---|---|
| **Manufacturing safety accident** | | | | | |
| Ollé-Goig [18] | 1987 | Spain | Rev.1 | 28/164 | Clinical, serological* |
| Pappas [19–21] | 2022 | China | A19† | 8/NR‡ | Serological, clinical, blood PCR |
| **Manufacturing exposure** | | | | | |
| Wallach [22] | 2008 | Argentina | S19 | 21/30 | Serological, clinical |
| Vives-Soto [23] | 2022 | Spain | Rev.1 (S19)§ | 47/115 | Serosurveillance¶ |
| Zhou [6] | 2022 | China | A19/S2 | 61/140 | Serosurveillance¶ |
| **Vaccine administration** | | | | | |
| Blasco [24] | 1993 | Spain | Rev.1 | 2/NR | Blood culture |
| Arapovic [25] | 2020 | Bosnia Herzegovina | Rev.1 | 1/NR | Blood cultured# |
| Vincent [26] | 1970 | France | S19 | 2/NR | Clinical |
| Nicoletti [27] | 1986 | USA | S19 | 1/NR | Clinical |
| Squarcione [28] | 1990 | Italy | Rev.1 | 1/NR | Clinical |
| Hatcher [29] | 2018 | USA | RB51 | 1/NR | Clinical |
| Pivnick [30] | 1966 | Canada | S19 | 21% | Clinical** |
| Stauffer [31] | 1998 | USA | RB51 | 4/32 | Clinical |
| Ashford [32] | 2004 | USA | RB51 | 19/26 | Clinical, blood culture†† |
| Avdikou [33] | 2005 | Greece | Rev.1 | 41/NR | Serological |
| Gunes [34] | 2013 | Turkey | Rev.1 | 10/46 | Serosurveillance‡‡ |
| Proch [35] | 2018 | India | S19 | 5/12 | Serological |
| Zhang [36] | 2018 | China | S2 | 51/206 | Serological, clinical |
| Pereira [37] | 2021 | Brazil | S19 / RB51 | 7/108 | Clinical§§ |

NR = not reported.

* Two positive cultures.

† A19 specific PCR assay in blood samples.

‡ General population atack rate 10,528 / 68,571 at november 2020. Aerosol transmission with some evidence of zoonotic transmission.

§ S19 vaccine manufacturing was marginal.

¶ No clinical infections.

# Confirmed by molecular methods (PCR).

** Reported through a questionnaire. Number of cases not available.

†† One positive culture taken from the inoculation site.

‡‡ Two clinical infections.

§§ Reported through an online questionnaire.

testing 68,571 people (attack rate 15.4%) [20]. There were no deaths reported related to this outbreak. Samples from the outbreak patients were subsequently analyzed using a specific PCR that identified the A19 strain, providing pathogenic evidence of the vaccine-derived infection outbreak [21]. Transmission appears to have been by aerosols spread by the wind in a southeast direction. The epidemiologic research reported the use of an expired disinfectant for cleaning that did not kill bacteria, coupled with a leak at the plant that allowed contaminated waste to be leaked in the air. Interestingly, the contamination affected some animals and there were cases of zoonotic spread to humans.

## Risk for vaccine manufacturing workers

Only three observational studies on vaccine manufacturing workers have been reported. In all of them, the diagnosis of *Brucella* infection was serological and the prevalence was related to the degree of exposure to the vaccine strains.

In Argentina, Wallach et al. (2008) evaluated the pathological consequences of exposure to the vaccine strain *Brucella abortus* S19 in 30 employees from vaccine manufacturing plants, between 1999 and 2006 [22]. Fifteen out of 21 laboratory employees with serologically-defined active infection showed clinical manifestations. Blood cultures were performed on nine patients and were negative in all cases. Fever, fatigue, joint stiffness, headache, muscle aches and neuropsycological symptoms were the most frequent findings. Only five of these workers recalled an accidental exposure, indicating that employees from laboratories producing the S19 vaccine are at risk of exposure to *Brucella abortus* by definition, and may become infected by this strain.

In 2022, a Spanish study by Vives-Soto et al. analyzed the human serologic response over time in a cohort of vaccine manufacturing workers exposed to the *Brucella melitensis* Rev.1 vaccine strain and, to a much lesser degree, the *Brucella abortus* S19 vaccine [23]. Although none of the workers developed symptomatic brucellosis, seropositivity was observed in 47 (40.9%) of the 115 individuals examined, indicating asymptomatic infection with the vaccine strains, despite strict safety measures. This seropositivity was significantly associated with greater level of proximity to *Brucella* vaccine strain cultures. Although serology does not allow to distinguish between vaccine and wild strains, the possibility of contact with *Brucella* outside the factory was minimal, given the almost absence of cases in the region where the factory is located: <0.08 cases per 100,000 inhabitants / year in the local official registry (https://www.sergas.es/Saude-publica/Documents/105/BEG_XXV-1.pdf). The fact that none of the workers studied developed the disease could be explained by the lower virulence of both vaccine strains and / or by a smaller bacterial inoculum due to strict compliance with safety measures. In Chongqing, China, Zhou et al. (2022) published a seroprevalence case-control study among employees of a vaccine factory engaged in the production of A19 and S2 *Brucella* strains with findings similar to those of the publication described above. They reported a sero-prevalence of 43.6% (61/140), although all workers were asymptomatic and no suspected or confirmed case was found [6]. The investigation pointed out that close contact with biological products and aerosols were the potential transmission routes in the context of insufficient personal protection and disinfection.

## Risk during vaccine administration

Publications of infection by vaccine strains during administration are scarce and heterogeneous, making unfeasible to carry out a meta-analysis. We have collected data from various types of publications: case reports, surveillance systems of occupational accidents, and investigations of outbreaks in vaccination campaigns. Three of these publications also address the effectiveness of security measures.

Firstly we summarized the six case report articles. In 1993, Blasco et al. published two cases of culture-positive *Brucella* infection in Spanish veterinarians accidentally exposed to the Rev.1 strain by needlestick [24]. In both individuals, the Rev.1 *Brucella mellitensis* strain was isolated from blood cultures. Later, in 2020, Arapovic reported the first case of Rev.1 human brucellosis in Bosnia and Herzegovina [25]. The patient, a farmer, had assisted the veterinarian in vaccinating his sheep, without wearing any personal protective equipment (PPE). The diagnosis was made by blood culture isolation of the Rev.1 strain, which was identified by molecular methods (multiplex PCR). The other 4 case reports had negative cultures and were diagnosed by clinical and serological criteria after accidental punctures with the *Brucella* vaccine strains S19 [26,27], Rev.1 [28] and RB51 [29].

The first report with an incidence rate was published in 1966 by Pivnick et al. based on a survey of Canadian veterinarians who vaccinated cattle with the S19 strain [30].They found

that 46% had accidentally injected themselves at least once and 45% of them developed moderate to severe symptoms (attack rate 20.7%). In the United States (USA), the RB51 vaccine strain replaced the S19 vaccine in 1996, since RB51 was found to be equally immunogenic but less virulent than S19 [12]. In 1998, the Centers for Disease Control and Prevention (CDC) published 32 notifications of unintentional inoculation or conjunctival exposure to the RB51 vaccine, occurring in Kansas (USA) in1997 [31]. Three of the cases reported inflammation at the inoculation site, and another person described systemic symptoms. Subsequently, Ashford et al. (2004) published findings from the CDC registry involving reports received from 26 veterinarians accidentally exposed to RB51 during animal vaccination in USA, between 1998 and 2002 [32]. Nineteen of them cited local or systemic symptoms, while 7 reported no adverse events associated with the accidental exposure. Only one of the veterinarians showed a positive culture, which was taken from the cutaneous injection site. Since current standard serologic assays cannot detect antibodies against RB51 infection [12], and the passive surveillance registry probably underestimates rates of needlestick injuries, the authors concluded that we cannot yet determine whether the RB51vaccine has the potential to cause systemic brucellosis in humans.

In 2005, Avdikou et al. described the results of a local brucellosis surveillance system implemented in a defined region of Northwestern Greece [33]. Of a total of 152 newly diagnosed cases recorded during a 2-year study period, 41 (27.0%) reported contact with the Rev.1 vaccine during its administration.

Gunes et al. published in 2013 a serosurveillance study of 46 veterinary staff assigned to a sheep vaccination campaign in Turkey using the Rev.1 strain [34]. Ten persons became seropositive (Rose Bengal test and Wright test ≥1/160), but only 2 developed symptoms of infection and were treated with antibiotics. At 6 months, all of them showed negative serology (Wright test). The study suffers from some limitations since serologic tests were not performed prior to the vaccination campaign, and the prevention measures applied were not stated.

Between 2015 and 2016, Proch et al conducted a study aimed to identify risk factors associated with occupational *Brucella* infection in 296 veterinary personnel in India [35]. Blood samples were taken from 279 individuals and the Rose Bengal, standard tube agglutination (STAT) and ELISA tests were performed. Previous *Brucella* needlestick injury with the S19 strain was reported in 12 individuals, of whom 5 had a positive serologic test for *Brucella*. After adjusting for other variables, the odds of having a positive serologic test were higher for non-veterinarians, individuals with more seniority and, paradoxically, for those using personal protective equipment (PPE). However, only 29/275 (10.5%) subjects used PPE, and the appropriateness of its use could not be assesed.

In 2017, an outbreak of brucellosis caused by S2 strain was reported in Tianzhu County, located in the Gansu province of China, during an animal vaccination campaign [36]. A total of 206 controllers participated in the immunization, of wich 51 were postive by serologic testing (infection rate: 24.8%). Although blood cultures of the 51 workers were negative, 48/51 (94.1%) suffered fatigue and sweat, 4 had fever, and 5 swelling of the testis. The vaccination work did not comply with biosafety recommendations, including improper handling in vaccination, inadequate use of PPE, and imperfect emergency measures.

Recently, in 2021, Pereira et al. published the results of an online questionnaire carried out on veterinarians registered to administer S19 and RB51 vaccines in Minas Gerais state, Brazil [37]. Three hundred and twenty-nine veterinarians were included in the analyses, using stratified random sampling. One hundred and eight (32.8%) of them cited accidental exposure to S19 or RB51 vaccine strains, 15 (4.6%) reported having had brucellosis, and 7 of those 15 considered that the infection was due to accidental exposure to *Brucella* vaccines. Poor knowledge of human brucellosis symptoms and lack of appropriate PPE use were risk factors for unintentional contact with S19 and RB51 vaccine strains.

## Discussion

Our results show that there is a risk of occupational infection by *Brucella* vaccine strains, although the small number of symptomatic infections recorded compared to the enormous number of doses administered [https://www.coherentmarketinsights.com/market-insight/brucellosis-vaccines-market-5038], suggests that most of them are subclinical. Humans can be infected through aerosol exposure and by mucosal and non-intact skin contact with live attenuated strains. Of the 5 vaccine strains currently used (Rev.1, S19, RB51, A19 and S2), Rev.1 seems to be the most virulent [7]. Prolonged contact with the pathogen, lack of information and instructions provided to the occupational groups exposed, and low adherence to personal protective equipment (PPE) in the work environment, appeared to be the main risk factors leading to infection by these vaccine strains [6,35–37]. Vaccine factories accidents are infrequent but can cause serious outbreaks due to aerial spread. On the other hand, despite strict protection measures, a percentage of vaccine manufacturing workers developed positive serology without symptomatic illness [6,23]. In fact, Buchanan et al. had observed in slaughterhouses that workers with positive serology showed a lower risk of acquiring brucellosis [38]. Among veterinarians and other vaccine administration workers, vaccine handling was the most reported source of exposure to *Brucella*.

The danger of these *Brucella* vaccine strains for use in human inoculation has been studied in various clinical trials. In 1962 Spink et al. carried out a clinical trial of the *Brucella* vaccines conducted in Minnesota, USA [7]; 11 (68.7%) of the 16 volunteers receiving the Rev.1 strain developed acute brucellosis, four of them requiring hospitalization, while only 4 (25.0%) of the 16 individuals receiving the S19 vaccine reported "undesirable sequelae". The efficacy and safety of human vaccination with *Brucella* attenuate strains, mainly S19, has been studied in some population studies in the last 60 years. Between 1952 and 1958, Vershilova et al. conducted the largest clinical trial on 3 million people engaged in the livestock/meat/food processing industries in the former Soviet Union, using the S19 strain for human vaccination [39]. They observed an 59.5% reduction in cases of human brucellosis, with a "high safety rate" for the vaccine. In 1992, in France, Strady et al. carried out a prospective phase IV study with the S19 strain, on 161 professionally exposed human volunteers [40]. The authors observed local pain after injection in 45.2% of subjects and systemic reactions in 5.0%; however, the clinical efficacy of the vaccine could not be evaluated due to an insufficient number of participants. Lastly, in 1994, Hadjichristodoulou et al. conducted a clinical trial on 271 volunteers in Greece; although the S19 vaccine caused some side effects in a quarter of subjects, it was considered safe enough for use on a large scale [41]. Nonetheless, there is currently no licensed anti-*Brucella* vaccine for humans. Having said that, at the present time, research is being conducted on developing safe, effective, cross-protecting, exclusively human vaccines due to *Brucella´*s zoonotic potential and possible use in bio-warfare [3, 4].

On the other hand, indirect exposure to *Brucella* vaccine strains has been reported, including the assistance to livestick births and abortions, handling dairy products, and analyzing contaminated samples in clinical laboratories. As explained below, symptomatic cases were not due to exposure during vaccine manufacturing or administration. In 1998, the CDC described the case of a stillborn calf, delivered by cesarean. The necropsy revealed that death was due to infection by the RB51 strain *Brucella abortus* [31]. The strain was isolated from placental and fetal lung tissue, as well as from the blood of the calf´s mother and was identified by molecular methods (PCR). The nine persons who participated in the procedures received post-exposure prophylaxis, and none developed brucellosis during the 6-month follow-up period. Besides, *Brucella abortus* RB51 and S19 is transiently excreted in the milk of vaccinated cattle and can survive throughout the manufacture and conservation processes of both fresh

and ripened cheeses [42] which poses a risk of infection through ingestion. In this setting, in 2015 Osman et al. reported a serosurveillance study on 100 asymptomatic farmworkers employed in Khartoum, Sudan [43]. Ten of them tested seropositive, including 4 milkers with a positive blood culture, and only in one of them the S19 strain was identified by PCR, but milk ingestion could not be excluded. Later, in 2018, Cossaboom et al. reported a case of human brucellosis associated with the consumption of unpasteurized cow's milk purchased from a dairy in Paradise, Texas, USA [44]. The CDC's Bacterial Special Pathogens Branch (BSPB) confirmed the isolate as *Brucella abortus* vaccine strain RB51. Recently, in 2021, Sarmiento-Clemente et al. reported in Houston, USA, the first case of neurobrucellosis due to the vaccine strain RB51 [45]. The patient was an 18-year old Hispanic female who had consumed unpasteurized cheese brought from Mexico 1 month before onset of symptoms. The strain was isolated in blood cultures and identified by molecular methods in the Houston Health Department/Laboratory. The authors also mentioned that the CDC reported 3 confirmed cases in the United States of human infection by RB51 through consumption of raw milk. Finally, in regard to clinical laboratory exposure, the CDC conducted in 2007 a laboratory proficiency test to a total of 254 laboratories of USA and Canada with potential RB51 exposure [46]. Despite correct labeling of the samples, 916 laboratory workers did not handled the RB51 samples properly, including 679 (74.1%) with high-risk exposures and 237 (25.9%) with low-risk exposures, although no cases of brucellosis were reported. The authors highlight the need for routine adherence to recommended biosafety practices when working with infectious organisms.

To protect workers from being infected by these live attenuated vaccines, all individuals exposed to *Brucella* vaccine strains should be considered in a high-risk category, and procedures should be implemented to minimize spills, splashes and aerosols, as well as accidental needlesticks. Appropriate PPE for vaccination or close exposure to vaccinated animals must include gloves, closed footware, eye protection, a face shield and respiratory protection [16]. During the vaccine manufacturing process and the handling of potentially contaminated laboratory samples, strict safety measures in compliance with the Biosafety in Microbiological and Biomedical Laboratories (BMBL) standards are currently in place. They include: a) at least class II Biological Safety Cabinets (BSCs), b) proper personal protective equipment (PPE) and c) use of primary and secondary barriers [47]. Following accidental exposure to *Brucella* vaccine strains, symptoms should be monitored and antibiotic post-exposure prophylaxis administered [16]. However, despite the exhaustive protection measures in place, a certain likelihood of exposure persists. Moreover, the applicability of these measures is not always guaranteed, due to human factors such as carelessness, negligence or malingering. Therefore, in addition to protocolize biosafety practices and adequate training for workers, programs monitoring adherence to these standards and quality audits should be implemented [46]. Besides, companies should actively monitor their employees through periodic check-ups that include *Brucella* serology as quality control of the measures applied.

The present review suffers from some limitations. Firstly, there are relatively few published studies on occupational exposure to *Brucella* vaccine strains. In addition, these studies are highly heterogeneous in terms of methodology and diagnostic criteria, as shown in Table 1. Moreover, many of them used serology as a diagnostic criterion, which does not distinguish infections caused by vaccine strains from those produced by wild strains. Only two of the studies in our analysis [35,37] evaluated which protective measures were the most effective for exposed workers (that is to say, proper use of PPE, careful handling of vials and needles, and correct disinfection measures), but they were not entirely conclusive due to the difficulty in assessing degree of individual adherence to PPE use.

In conclusion, brucellosis vaccines pose risk of contagion to humans, both from the manufacturing process and their administration to cattle. Recommended protective measures, such as proper use of PPE, employment of primary and secondary barriers and the handling of vaccines in Biological Safety Cabinets, appeared to reduce this risk. Nonetheless, evidence for the efficacy of these measures is weak, and the incidence of human infection with these vaccines is unclear. Therefore, it would be advisable to carry out observational studies and/or systematic registries using solid diagnostic criteria in populations exposed to *Brucella* vaccine strains. In addition, clinical and serological follow-up programs for exposed workers should be established.

## Acknowledgments

The authors thank Alexandra Salewski Msc for expert English review of the manuscript, and Carlos de Cabo PhD for preparation of the edition and presentation of the work for its publication.

## Author Contributions

**Conceptualization:** Esteban Rodríguez-Sánchez, Javier Solera.

**Data curation:** Manuel Vives-Soto, Amparo Puerta-García, José-Luis Pereira.

**Formal analysis:** Manuel Vives-Soto, Amparo Puerta-García.

**Investigation:** Manuel Vives-Soto, Javier Solera.

**Methodology:** Manuel Vives-Soto.

**Supervision:** Esteban Rodríguez-Sánchez, Javier Solera.

**Validation:** José-Luis Pereira.

**Writing – original draft:** Manuel Vives-Soto, Javier Solera.

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
