## [Decision Letter · Decision Letter 0]

28 Aug 2023

Dear Dr. Solera Santos,

Thank you very much for submitting your manuscript "What risk do Brucella vaccines pose to humans? A systematic review of the scientific literature on occupational exposure." for consideration at PLOS Neglected Tropical Diseases. As with all papers reviewed by the journal, your manuscript was reviewed by members of the editorial board and by several independent reviewers. In light of the reviews (below this email), we would like to invite the resubmission of a significantly-revised version that takes into account the reviewers' comments. 

Please respond to author queries, or explain why any changes will be not made

We cannot make any decision about publication until we have seen the revised manuscript and your response to the reviewers' comments. Your revised manuscript is also likely to be sent to reviewers for further evaluation.

Sincerely,

Georgios Pappas

Academic Editor

Ana LTO Nascimento

Section Editor

Please respond to author queries, or explain why any changes will be not made

Reviewer's Responses to Questions

**Key Review Criteria Required for Acceptance?**

**Methods**

-Are the objectives of the study clearly articulated with a clear testable hypothesis stated?

-Is the study design appropriate to address the stated objectives?

-Is the population clearly described and appropriate for the hypothesis being tested?

-Is the sample size sufficient to ensure adequate power to address the hypothesis being tested?

-Were correct statistical analysis used to support conclusions?

-Are there concerns about ethical or regulatory requirements being met?

Reviewer #1: The methods used for the systematic review are correct.

Reviewer #2: I have copied my comments under Summary and General Comments

Reviewer #3: The objective should include the risk of brucellosis due to milk consumption contaminated with vaccine isolate and the mechanism (microbiological and immunological).

The study design is not appropriate to capture all the vaccine related incidence, as such events may not be published in a scientific peer-review journals. Background vaccine use information is necessary but lacking.

The number of articles is small.

**Results**

-Does the analysis presented match the analysis plan?

-Are the results clearly and completely presented?

-Are the figures (Tables, Images) of sufficient quality for clarity?

Reviewer #1: (No Response)

Reviewer #2: I have copied my comments under Summary and General Comments

Reviewer #3: The results failed to show all the evidence of vaccine related brucellosis infections.

Table itself showed that the results are not representative of global vaccine related incidence. And they are not comparable or compared.

**Conclusions**

-Are the conclusions supported by the data presented?

-Are the limitations of analysis clearly described?

-Do the authors discuss how these data can be helpful to advance our understanding of the topic under study?

-Is public health relevance addressed?

Reviewer #1: The conclusions are in general supported by the data presented, although some specific comments are provided below.

Reviewer #2: I have copied my comments under Summary and General Comments

Reviewer #3: No.

**Editorial and Data Presentation Modifications?**

Reviewer #1: (No Response)

Reviewer #2: I have copied my comments under Summary and General Comments

Reviewer #3: (No Response)

**Summary and General Comments**

Reviewer #1: This is an important review on the pathogenicity for humans of Brucella vaccine strains used in the animal vaccination programs in endemic countries. Some specific comments to help to improve the manuscript are provided below:

1) Lines 34-36. “Despite strict protection measures, a percentage of vaccine manufacturing workers developed a positive serology to the vaccine strain, which may have conferred immunity.” 

COMMENT: Serological tests do not allow determining the Brucella species that elicited the antibody response. Moreover, cross-reactivity with other bacteria may lead to false positive results. In addition, please provide references supporting the protection conferred to humans by the accidental exposure to vaccine strains.

2) While a systematic review of the relevant literature has been performed, some reports of human infection by vaccine Brucella strains seem to be missing. For example, an important outbreak that took place in China in 2019 should be mentioned (Pappas G, Clin Infect Dis. 2022 Nov 14;75(10):1845-1847; Baoshan L et al., Transbound Emerg Dis 2021 Mar;68(2):368-374.) We also recommend including relevant information provided in chapter 8 of the book Brucellosis Clinical and Laboratory Aspects (Young EJ, Corbe MJ, CRC Press, 1989).

3) Lines 31-32. In our opinion, exposure accidents during vaccine manufacturing should not be considered exceptional, as several cases have been described.

4) Lines 41-44. “Human brucellosis is a zoonosis with worldwide distribution. Although the number of cases of brucellosis may be decreasing in the world, the continued presence of the disease in some endemic areas and the potential use of Brucella species as an agent of bioterrorism, make brucellosis a major public health hazard with important sanitary and economic repercussions”.

COMMENT: As the apparent decrease of brucellosis cases may be due to subnotification, I suggest to use “reported cases” instead of “cases”.

5) Lines 52-55. “However, these vaccines are not used in humans due to the high risk of developing acute brucellosis; they are capable of infecting humans with occupational exposure via the oral, nasal, conjunctival or genital routes, and by accidental needle inoculation”.

COMMENT: To our best knowledge, there are no reports of vaccine strain infection in humans through the genital route.

Reviewer #2: In this study, the authors reviewed the risk of occupational exposure to the Brucellosis vaccine. The research question is valid and is of interest to the journal audience. However, the study does not appear to be well designed and conducted.

Although the authors report that they followed the PRISMA guidelines, it is surprising to note that 11 of the 12 studies finally selected for inclusion in the systematic review were not in the systematic search they conducted but rather were selected based on a manual search of references found in the full-text articles reviewed. This nullifies any of the benefits of the systematic search and incorporates bias into the search that the systematic review aims to avoid. 

Secondly, the authors have a tendency to summarise information from the studies rather than comparing and contrasting or evaluating evidence and developing an argument. This is not acceptable, even in a simple critical review. 

Thus, although the subject is worthy of investigation, the manuscript would have to be substantially improved before it can be published in PLOS Neglected Tropical Diseases.

Reviewer #3: General comments

 At the first look at the first half of the title, I thought the scope is to discuss the risk from milk shed by vaccinated animals. The risk from milk cannot be ignored when considered Brucella vaccines. Also, microbiological and immunological aspects should be included. The number of articles examined is rather small, which may suggest the low attention paid by brucellosis endemic countries. The approach of systematic review is just summarizing these literatures, and there is no original analysis conducted for meaningful summary or comparisons. Moreover, I remember that a large-scale outbreak of brucellosis associated with a vaccine production unit in China a few years ago, but it is not included. Such incidence may not be in a scientific literature, and due to the same reason, the authors might have missed several other such accidents/incidence.

 In brucellosis endemic countries, as occupational exposure is always one of the most important risk factors, infections with wildtype Brucella are very common; which means the authors’ argument that sero-positivity is due to Brucella vaccine is flawed, though some or a few of them can be truly due to the exposure to the vaccine strain. The study lacked the investigation into the quantitative information on the actual use (relative usage) of different types of the vaccine and the cases. The authors argue that Rev1 is the most virulent vaccine but the quantitative or qualitative evidence is not enough and the statement is not convincing. The suitability of this article to PLoS NTD should be judged by the editor, but I am not convinced due to above reasons. However, the information on the risk of brucellosis vaccine to humans is still indeed scarce. I would like to encourage the authors to anyway improve the article to make it more informative.

Specific comments

Material and methods

Line 69: Please Italicize “Brucella”.

Results

Line 75-76: The sentence is not clear: “Eleven of these articles were identified by manual search of the references found in the full text articles reviewed, and one of them was included in the study”. Out of 12, 11 were found from manual search and 1 was by what? Manual search is also not clear. 

Line 108: “Since current standard serologic assays cannot detect antibodies against RB51 infection”, please provide the reference here. What about animals?

Line 146: RB51?

Line 158: Having brucellosis – having had brucellosis, or was the prevalence 4.6%?

Line 169: Sero-positivity associated with higher levels of exposure to the vaccine strains: what is the level of exposure you mean? High level is the high degree of invasion?

Discussion

Line 177: How did you judge that Rev.1 seems to be the most virulent?

Line 183: Please use references to state about the incidence in manufacturing workers.

Line 190: which country Spink conducted the clinical trial?

Line 199: clinical trial in Greece – it was considered safe enough: please discuss why it was considered so by the authors at that time.

Line 230: Please provide the key findings clearly when you refer systematic reviews. What were the risk factors for example? Exposure to vaccine strains through vaccination to animals?

Line 239-241: Enormous number of vaccine doses administered to humans? In which countries? Please present the evidence.

P266: Here please remind which were effective measures.

PLOS authors have the option to publish the peer review history of their article (what does this mean?). If published, this will include your full peer review and any attached files.

Reviewer #1: Yes: Jorge C. Wallach

Reviewer #2: No

Reviewer #3: No
---

## [Decision Letter · Decision Letter 1]

11 Dec 2023

Dear Dr. Santos,

Thank you very much for submitting your manuscript "What risk do Brucella vaccines pose to humans? A systematic review of the scientific literature on occupational exposure." for consideration at PLOS Neglected Tropical Diseases. As with all papers reviewed by the journal, your manuscript was reviewed by members of the editorial board and by several independent reviewers. The reviewers appreciated the attention to an important topic. Based on the reviews, we are likely to accept this manuscript for publication, providing that you modify the manuscript according to the review recommendations. 

please do these minor corrections requested by the reviewer before final acceptance

Sincerely,

Georgios Pappas

Academic Editor

Ana LTO Nascimento

Section Editor

please do these minor corrections requested by the reviewer before final acceptance

Reviewer's Responses to Questions

**Key Review Criteria Required for Acceptance?**

**Methods**

-Are the objectives of the study clearly articulated with a clear testable hypothesis stated?

-Is the study design appropriate to address the stated objectives?

-Is the population clearly described and appropriate for the hypothesis being tested?

-Is the sample size sufficient to ensure adequate power to address the hypothesis being tested?

-Were correct statistical analysis used to support conclusions?

-Are there concerns about ethical or regulatory requirements being met?

Reviewer #3: -Are the objectives of the study clearly articulated with a clear testable hypothesis stated? Yes

-Is the study design appropriate to address the stated objectives? Yes

-Is the population clearly described and appropriate for the hypothesis being tested? Not applicable

-Is the sample size sufficient to ensure adequate power to address the hypothesis being tested? Not applicable

-Were correct statistical analysis used to support conclusions? Not applicable

-Are there concerns about ethical or regulatory requirements being met? No

**Results**

-Does the analysis presented match the analysis plan?

-Are the results clearly and completely presented?

-Are the figures (Tables, Images) of sufficient quality for clarity?

Reviewer #3: -Does the analysis presented match the analysis plan? Yes

-Are the results clearly and completely presented? Yes

-Are the figures (Tables, Images) of sufficient quality for clarity? Yes

**Conclusions**

-Are the conclusions supported by the data presented?

-Are the limitations of analysis clearly described?

-Do the authors discuss how these data can be helpful to advance our understanding of the topic under study?

-Is public health relevance addressed?

Reviewer #3: -Are the conclusions supported by the data presented? Yes

-Are the limitations of analysis clearly described? Yes

-Do the authors discuss how these data can be helpful to advance our understanding of the topic under study? Yes

-Is public health relevance addressed? Yes

**Editorial and Data Presentation Modifications?**

Reviewer #3: Abstract

Line 40: risk factors – please rephrase unless statistics are conducted. For example, commonly associated with …

Line 44: separate ‘workersappeared’.

Introduction

Line 72: After the first appearance of Brucella, it can be abbreviated as B.

Result

Line 153, please show the date or month of the accident. In the line 157, it says ‘until November’, and readers would want to know how long after the accident the investigation was carried out.

Lines 154, 195, 215 and so forth: Please Italicize Brucella.

Line 196: one digit below zero – please unify this format throughout the manuscript. It should be ‘sero-positive rate’ or ‘sero-prevalence’, instead ‘infection rate’.

**Summary and General Comments**

Reviewer #3: My previous comments were addressed.

PLOS authors have the option to publish the peer review history of their article (what does this mean?). If published, this will include your full peer review and any attached files.

Reviewer #3: Yes: Kohei Makita

Figure Files:

Data Requirements:

Reproducibility:

References

---

## [Editor Report · Decision Letter 2]

27 Dec 2023

Dear Dr. Santos,

We are pleased to inform you that your manuscript 'What risk do Brucella vaccines pose to humans? A systematic review of the scientific literature on occupational exposure.' has been provisionally accepted for publication in PLOS Neglected Tropical Diseases.

Best regards,

Georgios Pappas

Academic Editor

Ana LTO Nascimento

Section Editor

=

<style type="text/css">p.p1 {margin: 0.0px 0.0px 0.0px 0.0px; line-height: 16.0px; font: 14.0px Arial; color: #323333; -webkit-text-stroke: #323333}span.s1 {font-kerning: none

</style>

---

## [Editor Report · Acceptance letter]

3 Jan 2024

Dear Doctor Solera Santos,

We are delighted to inform you that your manuscript, "What risk do Brucella vaccines pose to humans? A systematic review of the scientific literature on occupational exposure.," has been formally accepted for publication in PLOS Neglected Tropical Diseases.

Best regards,

Shaden Kamhawi

co-Editor-in-Chief

Paul Brindley

co-Editor-in-Chief
